# Insecticide Resistance in *Aedes aegypti* Mosquitoes: Possible Detection of *kdr* F1534C, S989P, and V1016G Triple Mutation in Benin, West Africa

**DOI:** 10.3390/insects15040295

**Published:** 2024-04-22

**Authors:** Tatchémè Filémon Tokponnon, Razaki Ossè, Sare Dabou Zoulkifilou, Gbenouga Amos, Houessinon Festus, Gounou Idayath, Aboubakar Sidick, Louisa A. Messenger, Martin Akogbeto

**Affiliations:** 1Ecole Polytechnique d’Abomey Calavi, Université d’Abomey-Calavi, Abomey-Calavi 01 BP 526, Benin; zoulkif72@gmail.com (S.D.Z.); amos1904@outlook.com (G.A.); houessinonfestus@gmail.com (H.F.); gounouidayathjoachelle@gmail.com (G.I.); 2Centre de Recherche Entomologique de Cotonou, Ministère de la Santé, Cotonou 06 BP 2604, Benin; ossraz@yahoo.fr (R.O.); sidick_aboubakar@yahoo.fr (A.S.); akogbetom@yahoo.fr (M.A.); 3Centre Béninois de la Recherche Scientifique et de l’Innovation (CBRSI), Cotonou BP 1665, Benin; 4Ecole de Gestion et d’Exploitation des Systèmes d’Elevage, Université Nationale d’Agriculture, Kétou BP 44, Benin; 5Department of Disease Control, London School of Hygiene & Tropical Medicine, London WC1E 7HT, UK; louisa.messenger@unlv.edu; 6Department of Environmental and Occupational Health, School of Public Health, University of Nevada, Las Vegas, NV 89154, USA

**Keywords:** *Aedes aegypti*, pyrethroid resistance, *kdr* mutations, detoxification enzymes, 10^ème^ arrondissement of Cotonou, Godomey-Togoudo, Benin

## Abstract

**Simple Summary:**

The effectiveness of chemical control of *Aedes (Ae.) aegypti* is threatened by the increasing frequency of insecticide resistance. This study aimed to determine, in two cities of Benin, the insecticide resistance profiles of *Ae. aegypti*, the presence of detoxification enzymes, and the frequency of *kdr* mutations. *Ae. aegypti* eggs were collected in the study areas using gravid *Aedes* traps (GATs). Centers for Disease Control and Prevention (CDC) bottle bioassays were used to assess the susceptibility status of adult female *Ae. aegypti,* followed by *kdr* screening using allele-specific PCR. The activity levels of key detoxification enzymes were measured among individual, unexposed, and un-engorged adult female *Ae. aegypti*. In both study sites, *Ae. aegypti* was resistant to the pyrethroids deltamethrin and permethrin, but susceptible to the carbamate bendiocarb. Significant over-expression of glutathione-*S*-transferases and under-expression of α and β esterases were observed in these vector populations. Three *kdr* mutations (F1534C, S989P, and V1016G) were possibly present in resistant *Ae. aegypti* at high frequencies, including the simultaneous occurrence of all three mutations in individual mosquitoes. Study findings will be used to inform prospective vector control strategies in Benin.

**Abstract:**

Epidemics of arboviruses in general, and dengue fever in particular, are an increasing threat in areas where *Aedes (Ae.) aegypti* is present. The effectiveness of chemical control of *Ae. aegypti* is jeopardized by the increasing frequency of insecticide resistance. The aim of this study was to determine the susceptibility status of *Ae. aegypti* to public health insecticides and assess the underlying mechanisms driving insecticide resistance. *Ae. aegypti* eggs were collected in two study sites in the vicinity of houses for two weeks using gravid *Aedes* traps (GATs). After rearing the mosquitoes to adulthood, female *Ae. aegypti* were exposed to diagnostic doses of permethrin, deltamethrin and bendiocarb, using Centers for Disease Control and Prevention (CDC) bottle bioassays. Unexposed, un-engorged female *Ae. aegypti* were tested individually for mixed-function oxidase (MFO), glutathione-*S*-transferase (GST) and α and β esterase activities. Finally, allele-specific PCR (AS-PCR) was used to detect possible *kdr* mutations (F1534C, S989P, and V1016G) in the voltage-gated sodium channel gene in insecticide-exposed *Ae. aegypti.* Most traps were oviposition positive; 93.2% and 97% of traps contained *Ae. aegypti* eggs in the 10^ème^ arrondissement of Cotonou and in Godomey-Togoudo, respectively. Insecticide bioassays detected resistance to permethrin and deltamethrin in both study sites and complete susceptibility to bendiocarb. By comparison to the insecticide-susceptible Rockefeller strain, field *Ae. aegypti* populations had significantly higher levels of GSTs and significantly lower levels of α and β esterases; there was no significant difference between levels of MFOs. AS-PCR genotyping revealed the possible presence of 3 *kdr* mutations (F1534C, S989P, and V1016G) at high frequencies; 80.9% (228/282) of the *Ae. aegypti* tested had at least 1 mutation, while the simultaneous presence of all 3 *kdr* mutations was identified in 13 resistant individuals. Study findings demonstrated phenotypic pyrethroid resistance, the over-expression of key detoxification enzymes, and the possible presence of several *kdr* mutations in *Ae. aegypti* populations, emphasizing the urgent need to implement vector control strategies targeting arbovirus vector species in Benin.

## 1. Introduction

*Aedes (Ae.) aegypti* mosquitoes are the main vector species of dengue viruses worldwide. The number of dengue fever cases reported to the World Health Organization (WHO) has increased dramatically by more than eightfold over the previous twenty years, from 505,430 cases in 2000 to over 2.4 million cases in 2010 and 5.2 million cases in 2019 [1]. *Ae. aegypti* is present in Benin and proliferating globally due to the development of trade, rapid urbanization, and high frequency of international travel [2]. Between 2010 and 2019, dengue fever cases have been diagnosed in Benin, resulting in at least one death [2].

In the absence of effective vaccines and available treatments, vector control remains the main strategy for dengue virus prevention. Vector control relies on the use of insecticides, such as pyrethroids, for house spraying and personal protection [3]. As a result of the strong selection pressures exerted using insecticides in agricultural practices and malaria control, including pyrethroid-only long-lasting insecticidal nets (LLINs) and indoor residual spraying (IRS) with pyrethroids, insecticide resistance among *Ae. aegypti* populations is common and widespread worldwide. According to the WHO, insecticide resistance is defined as “the ability of mosquitoes to survive exposure to a standard dose of insecticide; this ability may result from physiological or behavioral adaptation” [4]. Two of the main mechanisms underlying insecticide resistance in mosquitoes are alterations in the insecticide target site, including knock-down resistance (*kdr*) mutations in the voltage-gated sodium channel (*vgsc*) gene and increased metabolic activity [5,6]. Metabolic resistance to pyrethroids may be mediated by glutathione-*S*-transferases (GSTs), esterases, and mixed-function oxidases (MFOs) [7,8,9].

*Kdr* mutations have been studied in *Ae. aegypti* extensively; these mutations confer cross-resistance to pyrethroids and DDT by altering the structure of the *vgsc*, thus decreasing binding affinity of target insecticides [10]. Several *kdr* mutations have been identified in *Ae. aegypti* populations worldwide, including V1016I, V410L, S989P, I1011V, V1016G, I1011M, and F1534C [10,11,12,13]. In Africa, F1534C, V1016I, V410L, and S989P have been associated with pyrethroid resistance in *Ae. aegypti.* The presence of *kdr* has been reported in countries near to Benin, such as Burkina Faso, Nigeria, and Ghana [14,15,16,17].

The emergence and re-emergence of dengue fever epidemics requires effective vector control responses, including monitoring of vector population insecticide susceptibility. To date, there is a considerable paucity of insecticide resistance information for *Ae. aegypti* populations in Benin.

## 2. Materials and Methods

### 2.1. Study Sites and Mosquito Sampling

Adult *Ae. aegypti* mosquitoes were reared from eggs collected over a 2-week period in the 10^ème^ arrondissement of Cotonou and in Godomey-Togoudo, Abomey-Calavi from 31 December 2021 (Figure 1). This collection period coincided with the dry season, which is a period characterized by a scarcity of *Aedes* breeding sites. During this sampling period, a total of 73 gravid *Aedes* traps were set in the 10^ème^ arrondissement of Cotonou and 74 in Godomey-Togoudo. The traps used are black plastic pots that can contain half a liter of water, in which wooden egg-laying supports have been immersed. The traps were hung on the box or wall with a nail and a wire. The egg carriers were removed from each trap after 7 days, eggs were counted and then hatched according to standard insectary rearing procedures for *Aedes* species. A dipper was used to visually count the number of larvae per bowl. Larval hatching rate was measured by dividing the number of larvae by the total number of immersed eggs ×100. Emergent adults were counted by aspiration. Adult emergence rate was measured by dividing the number of emerged adults by the total number of larvae ×100.

### 2.2. Aedes Morphological Identification

Adult *Ae. aegypti* for insecticide resistance bioassays were identified using Fontenille’s taxonomic keys [18]. *Ae. aegypti* and other *Aedes* were distinguished by characteristic white stripes on their legs. Then, the thorax was used to differentiate the two species; *Ae. aegypti* has two thin white median lines with a lyre pattern, whereas *Ae. albopictus* has only one distinct white central line.

### 2.3. Insecticide Resistance Bioassays

Insecticide resistance profiles for *Ae. aegypti* field populations were measured using Centers for Disease Control and Prevention (CDC) bottle bioassays [19]. Two-to-five-day-old female *Ae. aegypti* mosquitoes were exposed to diagnostic doses of deltamethrin (10 µg/bottle), permethrin (15 µg/bottle), and bendiocarb (12.5 µg/bottle) in Wheaton 250 mL bottles, alongside an acetone-treated control bottle. In each bioassay, 10–25 unfed mosquitoes were introduced into each bottle and knock-down was scored every 15 min until all were dead, or up to two hours had elapsed; data were reported for the diagnostic time of 30 min for all insecticides. Approximately 100 mosquitoes (in 3–4 bottle bioassays) were tested per insecticide dose.

### 2.4. Measurement of Detoxification Enzyme Activity

Biochemical tests were performed to quantify the activity of families of detoxification enzymes, including non-specific esterases (α and β esterases), mixed-function oxidases (MFOs), glutathione *S*-transferases (GSTs) and total proteins in 3–5-day old female *Ae. aegypti* mosquitoes; a total of 80 non-insecticide exposed mosquitoes were tested per study site, in parallel with the Rockfeller susceptible strain of *Ae. aegypti* as a control [20].

Individual mosquitoes were ground in 200 µL of distilled water using sterile pestles, after which the lysate was centrifuged at 14,000 rpm for two minutes. For non-specific esterases, we used 2 substrates: α-naphthyl acetate and β-naphthyl acetate. Each plate was calibrated with the products of the esterase-substrate reaction (α-naphthyl acetate or β-naphthyl acetate). The reaction consisted of hydrolysis of α-naphthyl (or β-naphthyl) acetate by esterase with the formation of α-(or β)-naphthol with Fast Garnett Salt. This assay involved determining the quantity of α-(or β)-naphthol formed as a function of time. For each esterase plate, 10 µL of mosquito lysate was added in duplicate. Then, the standard curves of α or β naphthol were generated with Gen5 software (version 2.0) to determine each well concentration. To each well, 10 µL of supernatant or standards and 90 µL of Phosphate Saline Buffer (PBS) pH 6.5 + 1% triton ×100 was added. The plate was then incubated at 25 °C for 10 min, after which 100 µL of working solution (500 µL α-naphthyl acetate or β-naphthyl (0.03 M) + 2.5 mL PBS buffer (pH mL 6.5) + 7 mL H_2_O), was added to each well and the plate was again incubated at 25 °C for 30 min. Finally, 100 µL of 3 mM of a solution of Fast Garnett Salt dissolved in 10 mL distilled water was added to each well and the plate was incubated at 25 °C for 10 min. Plate absorbance values were read on an ELx808 spectrophotometer at 550 nm. The standard curve fitted a straight line: DO = ax + b where x is the quantity of α- or β-naphthol in the well. Using this formula with the optical density (OD) of the sample we calculated the amount of α- or β-naphthol produced by 10 µL of mosquito supernatant for 30 min. Esterase activity of each mosquito was expressed as µmol α- or β-naphthol produced/min/mg protein and calculated according to the following: α- or β-naphthol in µmol per mL/amount of protein in mg per mL/30.

For MFOs, 20 µL of mosquito lysate was added to 2 wells (technical duplicate) of a 96-well ELISA plate. Then, the standards curve of cytochrome C was generated with Gen5 software to determine the each well concentration. The first 2 columns are reserved for the H_2_O control and the standard range. For each duplicate of 20 µL of supernatant, 80 µL of 0.0625M Potassium Phosphate Buffer (KHPO_4_; pH 7.2) was added; then, 200 µL of working solution (1.6 mM of 3,3,5,5-tetramethylbenzidine dihydrochloride was dissolved in 5 mL ethanol and 15 mL of 0.25 M sodium acetate buffer pH 5.0). After adding 25 µL of 3% hydrogen peroxide, the plate was incubated for 30 min, and plate absorbance values were read on an ELx808 spectrophotometer at 630 nm. The standard curve was fitted to a second-degree curve: OD = ax2 +bx + c, where x is the amount of P450 equivalent units in the homogenates. Using this formula with the ODs of the samples, we calculated the quantity of “P450 equivalent units” (in nmol) produced by 20 µL of supernatant. This transformation was usually performed automatically by the software supplied with the spectrophotometer. The oxidase activity of each mosquito in nmol P450 equivalent units/mg protein is as follows: (nmol P450 equivalent units for 20 µL supernatant/2 mg protein in 10 µL supernatant).

For GSTs, to each duplicate of 10 µL supernatant we added 200 µL of working solution: (10 mM GSH prepared in 0.1 M phosphate buffer, pH 6.5-, and 3-mM Di chloronitrobenzene dissolve in methanol). For each microtiter plate, 2 blanks with 10 µL of distilled water + 200 µL of working solution was added and the absorbance was measured at 340 nm for 5 min. The GST activity for individual mosquitos was calculated as μmol GSH conjugated/min/mg protein using known extinction coefficients: (Milli DO × 0.21)/(5.76 × 1000)/(amount of protein in mg in 10 µL of supernatant.

For total protein, 10 µL of supernatant was added to 2 wells (technical duplicate) of 96-well ELISA plate. Then, the standards curve of Bovine Serum Albumin (BSA) was generated with Gen5 software to determine the each well concentration. We added 200 µL of working solution (19 mL bicinchoninic Acid Solution + 380 mL Copper (II) Sulfate Solution) as described by the manufacturer. The plate was incubated for 30 min at room temperature, after which the OD was read as an end point at 595 nm.

### 2.5. Ae. aegypti kdr Genotyping

After grinding each mosquito in 200 μL of 2% CTAB, samples were then incubated at 65 °C for 5 min. Thereafter, 200 μL of chloroform was added and each sample was centrifuged at 12,000 rpm for 5 min at room temperature after mixing by inversion at least 10 times. The supernatant was mixed with 200 μL of isopropanol and centrifuged at 12,000 rpm at room temperature for 10 min. The DNA pellet was washed in 200 μL of 70% ethanol by centrifugation for 5 min at 12,000 rpm. The DNA pellet was dried at room temperature and re-suspended in 140 μL of sterile H_2_O.

An allele-specific PCR (AS-PCR) was used to detect the presence of S989P, V1016G, and F1534C *kdr* mutations [12]. Each individual mosquito was tested by AS-PCR twice, the first PCR used a primer specific to the susceptible wild-type and the second PCR used a primer specific to the mutant. The primers used for the genotyping were the following: S989PF:5′AATGATATTAACAAAATTGCGC3′ and S989PR:5′GCACGCCTCTAATATTGATGC; V1016GF:5′GCCACCGTAGTGATAGGAAATC3′ and V1016GVal-R:5′CGGGTTAAGTTTCGTTTAGTAGC3′; and F1534CF:5′GGAGAACTACACGTGGGAGAAC3′and F1534CR:5′CGCCACTGAAATTGAGAATAGC3′.

Each reaction was carried out in a final volume of 25 µL containing 1X Eurogentec Taq buffer (Kaneka Eurogentec, Liège, Belgium), 1.5mM MgCl_2_, 400 µm of each dNTP 25 pmol forward primer, 25 pmol reverse primer, 25 pmol sensitive or mutant primer, and 1 unit of Eurogentec Taq polymerase (Kaneka Eurogentec, Liège, Belgium). PCR conditions were as follows: 1 cycle at 94 °C for 3 min, then 35 cycles of 94 °C for 30 s, 60 °C (for *F1534C* and *V1016G*) or 62 °C (for *S989P*) for 30 s and 72 °C for 1 min, followed by one cycle at 72 °C for 7 min.

PCR products were separated by electrophoresis in a 2% agarose gel, stained with ethidium bromide. PCR amplicon sizes for the detection of *kdr* mutations were 240 bp (S989P), 284 bp (F1534C), or 348 bp (V1016G). Non-allele-specific external primers produced bands of 594 bp (S989P), 517 bp (F1534C), or 592 bp (V1016G). No-template controls (NTCs) were run in parallel for all assays as negative controls. Laboratory susceptible specimens (Rockfeller) were used as positive controls.

### 2.6. Data Analysis

Global Positioning System (GPS) coordinates of the gravid *Aedes* traps were recorded using the OSM Tracker for Android application. Insecticide susceptibility test results were recorded and analyzed using Microsoft Excel 2016. All biochemical data were saved after reading on a computer connected to the plate reader. Transformations from the various right-hand equations to determine the quantity of the final product were performed automatically by the software (GeneS.1) supplied with the spectrophotometer. Statistical analyses were also performed using GraphPad-Prism 5 software (version 5.00, San Diego, CA, USA). The Mann–Whitney test was chosen for comparison between Rockefeller (susceptible strain), and field mosquitoes. Statistical significance was determined if *p* < 0.05. Statistical analyses were conducted in Stata/SE 17.0, including Pearson’s Chi-squared test to investigate deviations from Hardy–Weinberg equilibrium.

## 3. Results

### 3.1. Mosquito Sampling

A total of 147 gravid *Aedes* traps were used in the 2 study areas, of which 142 were positive (i.e., female *Aedes* oviposited in them); yielding an attractiveness rate of 97%. The total number of eggs obtained was 15,844; 8846 eggs in the 10^ème^ arrondissement of Cotonou, and 6998 eggs in Godomey-Togoudo.

Of the 8846 eggs obtained in the 10^ème^ arrondissement of Cotonou, 3154 hatched (35.7%) and 1240 adult *Aedes* mosquitoes emerged (39.3%). Of the 6998 eggs obtained in Godomey-Togoudo, 4918 eggs hatched (70.3%) and 2486 adult *Aedes* mosquitoes emerged (50.5%).

Of the 3726 total mosquitoes that emerged, 3723 were *Ae. aegypti* (>99%), while 3 were *Ae. albopictus* (<1%). One of the three *Ae. albopictus* came from the 10^ème^ arrondissement of Cotonou, and the others from Godomey-Togoudo.

### 3.2. Insecticide Resistance Profiles

Resistance to permethrin and deltamethrin was evident in both populations of *Ae. aegypti.* Mosquito mortality was 85.7% and 82.7%, following exposure to the diagnostic dose of deltamethrin or permethrin after 30 min, respectively, in the 10^ème^ arrondissement of Cotonou (Figure 2). Similarly, in Godomey-Togoudo, mortality was 87.9% and 88.8% with deltamethrin and permethrin, respectively. By comparison, complete susceptibility to bendiocarb was observed in both vector populations (100% mortality) (Figure 2).

### 3.3. Expression of Detoxification Enzymes

Field insecticide-resistant *Ae. aegypti* populations had significantly higher median levels of GSTs compared to the insecticide-susceptible Rockefeller reference strain in Godomey-Togoudo (*p* = 0.0003) and in the 10^ème^ arrondissement of Cotonou (*p* = 0.0046) (Figure 3A). By comparison, field *Ae. aegypti* did not differ in their expression levels of MFOs, compared to the susceptible strain (Figure 3B) and had significantly lower median levels of non-specific esterases (α and β esterases) (Figure 3C,D).

### 3.4. kdr Mutation Screening

Three *kdr* mutations (S989P, V1016G, and F1534C) were identified in 82.9% (262/316) of pyrethroid-resistant *Ae. aegypti* from both study sites (Table 1). Among bioassay survivors, S989P was present in the highest frequencies (0.88 to 0.92). This mutation was under significant selection in both the 10^ème^ arrondissement of Cotonou and Godomey-Togoudo (χ^2^ = 4.32; *p* = 0.038; and χ^2^ = 4.40; *p* = 0.036; Table 1). F1534C ranged in frequency from 0.69 to 0.70 in *Ae. aegypti* which survived pyrethroid exposure, with significant deviations from the Hardy–Weinberg equilibrium in both study sites (χ^2^ = 1.61; *p* = 0.033 and χ^2^ = 4.62; *p* = 0.0024, in the 10^ème^ arrondissement of Cotonou and Godomey-Togoudo, respectively). V1016G ranged in frequency from 0.62 to 0.73, with no evidence for ongoing selection in either site (Table 1). We identified 13 insecticide-resistant *Ae. aegypti* with the simultaneous presence of all 3 *kdr* mutations.

## 4. Discussion

To deploy appropriate control strategies targeting arbovirus vectors, it is crucial to understand the distribution of key mosquito species, their bionomics and insecticide resistance profiles. Study findings report *Ae. aegypti* in the 10^ème^ arrondissement of Cotonou and Godomey-Togoudo during the dry season, which might be explained by climate change, transport development, and increasing urbanization. Both populations of *Ae. aegypti* were characterized by pyrethroid resistance (to deltamethrin and permethrin), but complete susceptibility to bendiocarb. Pyrethroid resistance in these populations is not expected, given that insecticide treatment specifically targeting *Ae. aegypti* in Benin is rare; rather, it may have been driven by several alternate factors. It is possible that insecticidal interventions used to control other vector species, e.g., LLINs, IRS, or household mosquito repellents targeting *Anopheles* vectors of malaria, may have exerted indirect selection pressure on *Ae. aegypti* for the evolution of pyrethroid resistance. Furthermore, water contamination caused by pesticides used for agricultural practices may have also played a role in the development of resistance.

Regarding insecticide resistance mechanisms, *Ae. aegypti* populations were characterized by over-expression of GSTs and a slight, but non-significant, increase in the activity of MFOs. It is likely that over-expression of both types of metabolic enzymes could confer pyrethroid resistance in these populations. Further investigation is warranted of molecular mechanisms involving point mutations in the GSTe2 gene, which may be contributing to pyrethroid resistance in *Ae. aegypti* in Benin. By comparison, under expression of non-specific esterases (α and β esterases) observed in these same populations may explain the strong sensitivity to bendiocarb; therefore, this insecticide has the potential to be used in both areas to suppress *Ae. aegypti* population densities.

Several mutations at nine different loci in *Ae. aegypti* have been identified, which are implicated in reduced insecticide susceptibility. Of these, F1534C, S989P, and V1016I are widely reported *kdr* mutations and have been associated with DDT and pyrethroid resistance [13,14,15,16]. In this study, AS-PCR genotyping revealed the presence of S989P, F1534C, and V1016G mutations in both populations of *Ae. aegypti* in Benin; 82.9% of genotyped mosquitoes carried at least one mutation. The frequencies of S989P in both study sites (63% and 78%) were much higher compared to that observed in Nigeria (7%) [16]. Paradoxically the high mutation frequencies identified in this study were accompanied by only a moderate amount of phenotypic pyrethroid resistance. It is possible that the high fitness cost associated with these mutations may be adversely impacting mosquito survival [21].

To our knowledge, this is the first report of V1016G in Africa, and this is the first time all three *kdr* mutations have been detected in *Ae. aegypti* in Benin. This presence of the V1016G gene mutation in West Africa represents a further expansion of the geographic range of this mutation and is an important finding. However, it would be necessary to confirm this mutation in positive samples using Sanger sequencing; this is a limitation of this study.

We identified 13 insecticide-resistant *Ae. aegypti* with the simultaneous presence of all three *kdr* mutations. The co-occurrence of two or three *kdr* mutations has been previously reported in China, Nigeria and Malaysia, which resulted in highly intense pyrethroid resistance [12,16,22,23]. Other *kdr* mutations, T1520I and F1534L, have also been found in *Ae. aegypti;* however, the S989P/V1016G/F1534C mutation is the most widespread, followed by S989P/V1016V and V1016G/F1534C. A quadruple mutation, S989P/V1016G/T1520I/F1534C, was also identified in 2022 in Myanmar in *Ae. aegypti* [24]. High prevalence of the F1534C mutation has been reported previously, but intense phenotypic resistance to pyrethroids is only found when accompanied by the *V1016I* mutation or V1016G, S989P, and/or V410L mutations [24,25]. Recent work has also highlighted specific profiles of metabolic gene overexpression associated with resistance to pyrethroids and organophosphates in *Ae. aegypti* popula-tions from two localities in Puerto Rico [26]. In this study, the detection of multiple *kdr* mutations and overexpressed detoxification enzymes in the major dengue virus vector *Ae. aegypti* is concerning and cautions against the use of pyrethroids in arbovirus control programs in Benin. In this part of West Africa, additional surveillance activities are needed to further investigate other co-occurring molecular and metabolic insecticide resistance mechanisms, as well as to assess the susceptibility of these *Ae. aegypti* populations to alternate pyrethroids (e.g., alpha-cypermethrin, cyfluthrin, or lambda-cyhalothrin), with or without synergists (e.g., piperonyl butoxide), or other classes of insecticides (e.g., organophosphates, neonicotinoids, pyrroles etc.) to identify efficacious insecticidal control measures. In addition to these vector control tools, information education communication and behavior change communication (IEC/BCC) initiatives are needed to encourage local community members to properly dispose of stagnant water sources, which act as potential *Aedes* breeding sites, and to promote the use personal protective measures (including the use of topical repellents and wearing of long clothing) to prevent mosquito bites. In the event of a dengue fever epidemic, the Ministry of Health in Benin is cautioned against relying exclusively on pyrethroids for vector control; combination and/or rotation of insecticides with different active ingredients may be required to suppress populations of pyrethroid-resistant *Ae. aegypti*.

## 5. Conclusions

This study reports resistance to deltamethrin and permethrin in *Ae. aegypti* populations, collected from the 10^ème^ arrondissement of Cotonou and Godomey-Togoudo in Abomey-Calavi. Molecular and metabolic mechanisms associated with pyrethroid resistance included the *kdr* mutations F1534C, S989P, and V1016G, and significant over-expression of certain detoxification enzymes. To our knowledge, this is the first report of V1016G in Africa, and this is the first time all three *kdr* mutations have been detected in *Ae. aegypti* in Benin, suggesting alternative vector tools may be required for arbovirus control in this part of West Africa. Study results highlight the importance of strengthening and scaling-up surveillance activities to respond to the control of vector-borne diseases.

## Figures and Tables

**Figure 1 insects-15-00295-f001:**
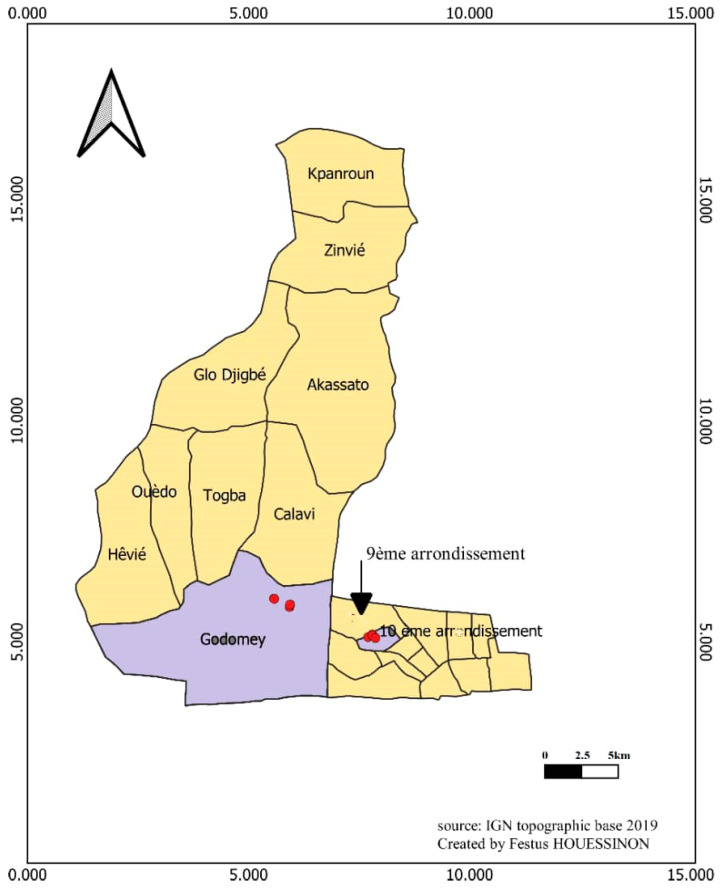
Map of trapping sites in Cotonou and Godomey-Togoudo, Benin. The dots correspond to different traps sites.

**Figure 2 insects-15-00295-f002:**
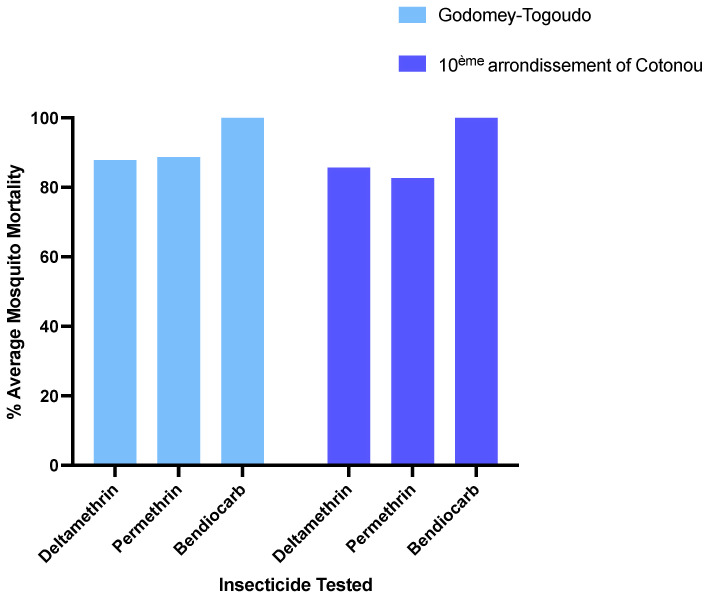
Susceptibility of *Ae. aegypti* to diagnostic doses of three insecticides in Godomey-Togoudo and the 10^ème^ arrondissement of Cotonou.

**Figure 3 insects-15-00295-f003:**
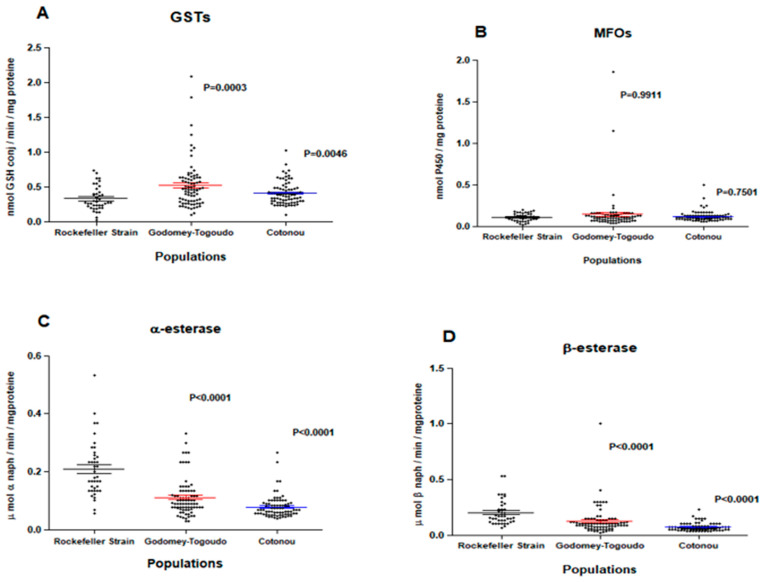
Expression levels of glutathione-*S*-transferases (**A**), mixed-function oxidases (**B**), α (**C**) and β (**D**) esterases among *Ae. aegypti* Rockefeller strain (insecticide-susceptible) and two insecticide-resistant *Ae. aegypti* field populations collected from Godomey-Togoudo and in the 10^ème^ arrondissement of Cotonou. Medians for each group are represented by solid-colored lines, accompanied by 95% confidence intervals.

**Table 1 insects-15-00295-t001:** *kdr* mutation (S989P, V1016G, and F1534C) allele frequencies in Benin.

*kdr* Mutation	Study Site	# Mosquitoes Tested	Homozygote Mutation (RR)		Heterozygote Mutation (RS)		Homozygote Wild Type (SS)		Allele Frequency				χ^2^ Test		*p*-Value	
			Dead	Alive	Dead	Alive	Dead	Alive	R		S					
									Dead	Alive	Dead	Alive	Dead	Alive	Dead	Alive
S989P	10^ème^ arrondissement of Cotonou	24	0	9	9	3	3	0	0.38	0.88	0.62	0.12	4.32	0.245	0.038	0.621
	Godomey-Togoudo	54	15	21	8	4	6	0	0.66	0.92	0.34	0.08	4.40	0.189	0.036	0.664
F1534C	10^ème^ arrondissement of Cotonou	78	33	15	12	9	5	4	0.78	0.70	0.22	0.30	4.52	1.61	0.033	0.204
	Godomey-Togoudo	111	31	23	21	12	17	7	0.60	0.69	0.40	0.31	9.20	4.62	0.0024	0.032
V1016G	10^ème^ arrondissement of Cotonou	33	5	7	5	7	6	3	0.47	0.62	0.53	0.38	2.22	0.279	0.136	0.597
	Godomey-Togoudo	16	1	8	1	3	1	2	0.50	0.73	0.50	0.27	0.333	2.22	0.564	0.136

#: number.

## Data Availability

Data are contained within the article.

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
