# Peer review of "Insecticide Resistance in Aedes aegypti Mosquitoes: Possible Detection of kdr F1534C, S989P, and V1016G Triple Mutation in Benin, West Africa"

_insects, 2024, doi:10.3390/insects15040295_

Round 1

Reviewer 1 Report (New Reviewer)

Comments and Suggestions for Authors

I have noticed that the manuscript has been revised to some extent. This article conducts an in-depth study on the mechanisms of resistance to pyrethroids in Aedes aegypti mosquitoes in the West African region of Benin, marking the first report of the V1016G mutation in this area and identifying the coexistence of kdr mutations (S989P, V1016G, and F1534C). This research is of significant importance for understanding the development of insecticide resistance in mosquitoes, especially in efforts within public health to control mosquito-borne diseases. The strengths of the paper lie in its high originality, the significant scientific and practical value of its content, as well as its rigorous experimental design and comprehensive data analysis, providing a scientific basis for resistance management and mosquito control strategies. Considering these factors, I believe this study will have a positive impact on research into mosquito resistance and public health practices.

Author Response

Thinks a lot Dear Revewers

Reviewer 2 Report (Previous Reviewer 2)

Comments and Suggestions for Authors

This manuscript examines phenotypic resistance to permethrin, deltamethrin, and bendiocarb in two populations of Aedes aegypti from Benin. Overall, the manuscript is useful and provides a good baseline for future studies. It is interesting work and the Authors examine a variety of mechanisms. Notably, only limited IR to the pyrethroids and full susceptibility to the carbamate was observed. The enzyme activity figure is quite well done and represents a lot of effort.

There are several critical issues that must be addressed to make sense of the paradox noted by the Authors. As noted, there is little pyrethroid IR but both elevated (or reduced) enzyme activity and extensive kdr mutation frequencies are detected. Several items need to be addressed (see specific line items below) but most of the major issues regard the kdr testing and whether the assay results are valid in the likely presence of unassessed SNPs (ex 1016I). Proper controls (and the results of these controls) are not described so it is hard to accept that the levels observed are correct. Without controls though, it is impossible to sort out what is present.

Additional context for these findings (If they are correct) must be provided. Why was 1016G assessed but 1016I was not? 1016I has been widely found in West Africa, 1016G has not. Does the presence of 1016I confound the AS-PCR for 1016G? How do these results fit into the context of previous kdr studies in neighboring countries in West Africa?

L3: Should kdr be italicized?

L83: Reference 15 (Kawada) is a study from Myanmar and thus not properly cited with African studies. It seems to fit better with the previous sentence.

Also suggest that you describe a bit more (a few sentences) what about kdr mutations near Benin. What is the current state of knowledge of kdr in countries around Benin?

L98-100: Both sentences describe a rate as the result but neither actually describe a rate but instead provide only an explanation of a count. Please describe how this becomes a "rate."

Also, it is unclear what measuring by an aspirator is exactly? Does this mean they were removed and counted by aspiration?

L106: The link for the CDC bioassay did not work so I was unable to answer a critical question. Were the AI DD and DT used the CDC recommended doses? The CDC CONUS-508 protocol recommends permethrin at 43ug/bottle. The previous CDC protocol used permethrin at 15ug/bottle.

If the doses used vary from the specified standard, how this may change the determination of resistant or susceptible should be clearly discussed.

L115: The text says Aedes spp. tested in IR bioassays... Does this mean that the Aedes tested in the bottle may have been mixed species? L105 says Ae. aegypti but this line calls this into question. Studies from the Americas have shown IR profiles of Ae. aegypti and Ae. albopictus are often vastly different. If tested as mixed populations, how will this change the results/conclusions?

Section 2.4: Very nice description of the methods used. A couple of important items are missing. First, what negative and positive controls were used (as in blank wells and product controls- alpha-napthol, beta-napthol, etc) in these assays?

Second, were control mosquitoes (as in Laboratory susceptibles) also used to establish a baseline? How do we know that levels are elevated in a population without a baseline?

Section 2.5: Two issues to address. 

The no template control is stated but what about controls with or wthout the specific alleles? Were known kdr negative strains used with this AS-PCR? Did they give the expected results? Without proper AS controls, it is difficult to accept that the results are valid.

The 1016I kdr mutation has been shown to be present in West Africa, the Americas and islands off the West of Africa. Here you assess 1016G which tends to be primarily present in Asia. Why 1016G and not 1016I or even better, why not assess both? How does this AS-PCR for 1016G handle the presence of 1016I? Does it amplify or not? Again, proper allele controls are critical.

L206: L207-209 shows 82%, 85%, 88% & 88% percent mortality at the DT for permethrin and deltamethrin. I do not think the use of "high resistance" is proper here. Nearly all were killed. More precise to say "Resistance at the DT to..."

Also, what was seen over the full 2 hr exposure period mentioned in the methods? How soon did you observe 100% mortality?

Also, it is unclear whether the lower doses here 10 ug/bottle) play a role. If permethrin DD had been at 43ug/bottle you would likley have seen no IR. Please see comment on method above.

Please state the result for the control bottles.

Section 3.3: Here the ROCK strain is used as a control strain. This strain is not mentioned in the methods.

Figure 3: Very nice figure.

Table 1: This would be much more approachable as a grouped column graph. The Chi squared results can be listed in the text as discussed.

I do have significant reservations about the validity of this Table as there is no mention of proper controls. Did the ROCK strain show as expected with none of these mutations? How does the locally present 1016I allele confound this AS-PCR?

L256: How do you justify the statement of "moderate-high" IR? Results of bioassay showed greater than 80% mortality at DT with a weak DD.

L281-284: Ultimately, it is impossible to determine that your (kdr) results are correct without the proper allele present and allele negative controls. Clearly, there is not much phenotypic IR to pyrethroids so the high frequency of particular kdr mutations that have been linked to strong IR in rigorous studies (see HH Chung work from Taiwan CDC, Chang et al 2009, Estep et al 2018, Mack et al 2021 as a few examples) is indeed paradoxical. It seems to point to another issue but without proper controls, what can we conclude? Not much.

Note also, that stating something like 82.9% had at least one kdr mutation is not very useful. As we have learned more about the phenotypic effect of these kdr mutations we have seen more and more support that these kdr mutations operate as ensembles to produce a strong effect and that genotype is more important (in Aedes aegypti) than just allele frequency (see refs above). The best example of the limited value of an isolated frequency is the study of Scott el al that showed 1534C alone was a pretty weak mutation. Studies from the Americas (Mexico, Peru, US) have all shown high levels of 1534C but only strong pyrethroid resistance when accompanied by 1016I.

L287: Note that 2 or 3 kdr mutations have been shown many times in the Americas including US, Mexico, Peru etc.

As noted at the beginning, there is much value to this study and it should be published once the critical deficits are addressed.

Comments on the Quality of English Language

Minor edits needed. A few are noted in the line comments above.

Author Response

This manuscript examines phenotypic resistance to permethrin, deltamethrin, and bendiocarb in two populations of Aedes aegypti from Benin. Overall, the manuscript is useful and provides a good baseline for future studies. It is interesting work and the Authors examine a variety of mechanisms. Notably, only limited IR to the pyrethroids and full susceptibility to the carbamate was observed. The enzyme activity figure is quite well done and represents a lot of effort.

There are several critical issues that must be addressed to make sense of the paradox noted by the Authors. As noted, there is little pyrethroid IR but both elevated (or reduced) enzyme activity and extensive kdr mutation frequencies are detected. Several items need to be addressed (see specific line items below) but most of the major issues regard the kdr testing and whether the assay results are valid in the likely presence of unassessed SNPs (ex 1016I). Proper controls (and the results of these controls) are not described so it is hard to accept that the levels observed are correct. Without controls though, it is impossible to sort out what is present.

Additional context for these findings (If they are correct) must be provided. Why was 1016G assessed but 1016I was not? 1016I has been widely found in West Africa, 1016G has not. Does the presence of 1016I confound the AS-PCR for 1016G? How do these results fit into the context of previous kdr studies in neighboring countries in West Africa?

In Africa, a number of kdr have been identified, and we have taken them into account in our introductory context. In our sub-region, the country whose context is closest to Benin's is Nigeria.  Recent work in Nigeria has reported the presence of F1534C and S989P, but an absence of V1016G. This justifies the fact that we chose to search for the F1534C, V1016G and S989P mutations, which are associated with pyrethroid resistance with greater effects due to their co-occurrence, whereas the presence of the V1016I mutation alone is not associated with pyrethroid resistance (Chen et al 2019).

L3: Should kdr be italicized?

Yes, kdr should be in Italics because it's linked to the gene. (Li et al 2015, Fagbohun et al 2022 et Sombié et al 2023). We have corrected this throughout the manuscript.

L83: Reference 15 (Kawada) is a study from Myanmar and thus not properly cited with African studies. It seems to fit better with the previous sentence.

Yes, observation was taken into account

Also suggest that you describe a bit more (a few sentences) what about kdr mutations near Benin. What is the current state of knowledge of kdr in countries around Benin?

Additional information was added to the introduction

L98-100: Both sentences describe a rate as the result but neither actually describe a rate but instead provide only an explanation of a count. Please describe how this becomes a "rate."

Yes, observation was taken into account

Also, it is unclear what measuring by an aspirator is exactly? Does this mean they were removed and counted by aspiration?

Emergent adults were by counted by aspiration. Adult emergence rate was measured by dividing the number of emerged adults by the total number of larvae ×100.

L106: The link for the CDC bioassay did not work so I was unable to answer a critical question. Were the AI DD and DT used the CDC recommended doses? The CDC CONUS-508 protocol recommends permethrin at 43ug/bottle. The previous CDC protocol used permethrin at 15ug/bottle.

Indeed, an error slipped through in the reporting, the protocol used 10ug of deltamethrin, 12.5 of bendiocarb and 15ug of pertmetrhine.

It is discussed in the text

If the doses used vary from the specified standard, how this may change the determination of resistant or susceptible should be clearly discussed.

It is discussed in the text

L115: The text says Aedes spp. tested in IR bioassays... Does this mean that the Aedes tested in the bottle may have been mixed species? L105 says Ae. aegypti but this line calls this into question. Studies from the Americas have shown IR profiles of Ae. aegypti and Ae. albopictus are often vastly different. If tested as mixed populations, how will this change the results/conclusions?

The susceptibility tests were poorly formulated. The tests were carried out only on morphologically identified Aedes aegypti. Correction was taken into the manuscript. We have switched sections, since the morphological identification was performed first, prior to bioassay testing.

Section 2.4: Very nice description of the methods used. A couple of important items are missing. First, what negative and positive controls were used (as in blank wells and product controls- alpha-napthol, beta-napthol, etc) in these assays?

The description of positive and negative controls is included in each section for detoxification enzymes.

Second, were control mosquitoes (as in Laboratory susceptibles) also used to establish a baseline? How do we know that levels are elevated in a population without a baseline?

The comparison was made with Rockefeller reference strains which were used

Section 2.5: Two issues to address. 

The no template control is stated but what about controls with or wthout the specific alleles? Were known kdr negative strains used with this AS-PCR? Did they give the expected results? Without proper AS controls, it is difficult to accept that the results are valid.

Template controls (NTCs) were run in parallel for all assays. Laboratory susceptibles specimens (Rockfeller) were used as positive controls and empty wells were used as negative controls.

The 1016I kdr mutation has been shown to be present in West Africa, the Americas and islands off the West of Africa. Here you assess 1016G which tends to be primarily present in Asia. Why 1016G and not 1016I or even better, why not assess both? How does this AS-PCR for 1016G handle the presence of 1016I? Does it amplify or not? Again, proper allele controls are critical.

Recent work in Nigeria has reported the presence of F1534C and S989P, but an absence of V1016G. This justifies the fact that we chose to search for the F1534C, V1016G and S989P mutations, which are associated with pyrethroid resistance with greater effects due to their co-occurrence, whereas the presence of the V1016I mutation alone is not associated with pyrethroid resistance (Chen et al 2019).

For PCR, we used primers specific to 1016G and not 1610I.

L206: L207-209 shows 82%, 85%, 88% & 88% percent mortality at the DT for permethrin and deltamethrin. I do not think the use of "high resistance" is proper here. Nearly all were killed. More precise to say "Resistance at the DT to..."

It is taken into account

Also, what was seen over the full 2 hr exposure period mentioned in the methods? How soon did you observe 100% mortality?

After 2 hours, the test was stopped, but 100% mortality for bendiocarb was observed well before 30 minutes.

Also, it is unclear whether the lower doses here 10 ug/bottle) play a role. If permethrin DD had been at 43ug/bottle you would likley have seen no IR. Please see comment on method above.

Please state the result for the control bottles.

Your concern has been taken into account in the correction of the active ingredient dose of 15ug/bottle of permethrin (see CDC Protocols cited above).

For the controls, it's because all the mosquitoes in the controls were alive and we didn't need to make any corrections.

Section 3.3: Here the ROCK strain is used as a control strain. This strain is not mentioned in the methods. It is ok

Figure 3: Very nice figure. ok

Table 1: This would be much more approachable as a grouped column graph. The Chi squared results can be listed in the text as discussed.

I do have significant reservations about the validity of this Table as there is no mention of proper controls. Did the ROCK strain show as expected with none of these mutations? How does the locally present 1016I allele confound this AS-PCR?

The format for tracking changes in a file that has been corrected several times is not easy to read, but once it has been accepted, it seems easier to read. The 1016I allele has not been the focus of our work, and ongoing research will provide further details.

L256: How do you justify the statement of "moderate-high" IR? Results of bioassay showed greater than 80% mortality at DT with a weak DD.

No moderate-high but resistance It is ok in the text

L281-284: Ultimately, it is impossible to determine that your (kdr) results are correct without the proper allele present and allele negative controls. Clearly, there is not much phenotypic IR to pyrethroids so the high frequency of particular kdr mutations that have been linked to strong IR in rigorous studies (see HH Chung work from Taiwan CDC, Chang et al 2009, Estep et al 2018, Mack et al 2021 as a few examples) is indeed paradoxical. It seems to point to another issue but without proper controls, what can we conclude? Not much.

We understand your concern, but the work was carried out with specific alleles, which confirms the controls used. Further work will take this into account, if possible using sequencing techniques.

Note also, that stating something like 82.9% had at least one kdr mutation is not very useful. As we have learned more about the phenotypic effect of these kdr mutations we have seen more and more support that these kdr mutations operate as ensembles to produce a strong effect and that genotype is more important (in Aedes aegypti) than just allele frequency (see refs above). The best example of the limited value of an isolated frequency is the study of Scott el al that showed 1534C alone was a pretty weak mutation. Studies from the Americas (Mexico, Peru, US) have all shown high levels of 1534C but only strong pyrethroid resistance when accompanied by 1016I.

Here we have not asserted but reported our findings. We have taken this concern into account in the discussion. High levels of the F1534C mutation have been observed, but only high resistance to pyrethroids is found in previous work when accompanied by the V1016I mutation or when accompanied by 1016G, S989P and/or V410L.

L287: Note that 2 or 3 kdr mutations have been shown many times in the Americas including US, Mexico, Peru etc.

Even 2 mutations in Nigeria and 3 Kdr mutations in Aedes aegypti in Burkina Faso in Africa

Reviewer 3 Report (New Reviewer)

Comments and Suggestions for Authors

The article “Insecticide resistance in Aedes aegypti mosquitoes: first evidence of kdr F1534C, S989P and V1016G triple mutation in Benin, West Africa” contributes to the knowledge of insecticide resistance in Aedes aegypti in West Africa, a topic worth to investigate since often vectors others than anopheline mosquitoes are neglected.

I would suggest the paper for publication but there is one aspect I would like to understand which is the dosages used for the bioassays; the reference is not anymore available and online different diagnostic dosages can be found. For permethrin for example I found a dosage of 15ug /bottle instead of 10 and such a difference could impact the results. I would suggest to check this twice and use a reference to a published document and not only a web-link.

For the rest there are only two minor questions:

-            When giving the PCR- protocols give for all reagents to concentrations and not only the volume you use in PCR

-            Line 283-284:  How could the possible high fitness cost of the found mutations explain the relatively low resistance level you found?

Author Response

I would suggest the paper for publication but there is one aspect I would like to understand which is the dosages used for the bioassays; the reference is not anymore available and online different diagnostic dosages can be found. For permethrin for example I found a dosage of 15ug /bottle instead of 10 and such a difference could impact the results. I would suggest to check this twice and use a reference to a published document and not only a web-link.

This was an error that has been corrected in the document.

For the rest there are only two minor questions:

-            When giving the PCR- protocols give for all reagents to concentrations and not only the volume you use in PCR

Concentrations are available in the document

-            Line 283-284:  How could the possible high fitness cost of the found mutations explain the relatively low resistance level you found?

A diversity of susceptibility levels has been observed in the area and further work is underway in this direction. 

Round 2

Reviewer 2 Report (Previous Reviewer 2)

Comments and Suggestions for Authors

This revision is improved and I appreciate the efforts undertaken to improve this manuscript. As noted at last review, most were minor and easily remedied. Only one major issue remains unresolved in this revision and respectfully, neither the response nor revision resolves the issue.

Demonstration of the presence of a 1016G in West Africa is a new expansion of the geographic range of 1016G and is an important finding. Since it is new and important, it requires a higher level of proof than just AS-PCR without adequate controls. At the minimum, there should be Sanger of some of these 1016G positive samples to verify the finding. While I understand that 1016I was not the goal of this study (though it certainly should have been considered since it is present in neighboring countries and your data shows that 1534C, it's partner in strong resistance, is present) it must be dealt with as a confounding factor by the inclusion of adequate controls. How do you know the AS-PCR method used doesn't give a false positive for 1016G when 1016I is present? Inclusion of the correct controls (for the 1016I and 1016G alleles) in the AS-PCR or a secondary method of confirmation would make me much more confident of the answer. 

This finding of 1016G in West Africa may well be true but the data here does not convince me without proper controls or alternatively, confirmation via a gold standard method like Sanger.

L206-207: ROCK is negative for all kdr SNPs. It is not a positive control with respect to kdr alleles. See Kandel et al 2019 as one example where the PR strain was included as a control for the 1534C allele.

Comments on the Quality of English Language

Only minor edits required.

Author Response

I thank the reviewer for this concern

This presence of the V1016G gene mutation in West Africa represents a further expansion of the geographic range of this mutation and is an important finding. However, it would be necessary to confirm this mutation in positive samples using Sanger sequencing. This is a limitation of this study and was taken into account in the discussion (lines 380-383)

Round 3

Reviewer 2 Report (Previous Reviewer 2)

Comments and Suggestions for Authors

Dear Authors,

As noted at the previous reviews, the manuscript is an interesting one and well written, but this version still suffers from a lack of proper positive controls that really confirm the central finding stated by the title. This finding is very impactful because it demonstrates an expansion of 1016G into a new area but also because it will require that future studies in the area must include testing for a broader range of possible alleles. It is puzzling to look for the presence of 1016G, which as noted, has not been found in the area, but not assess 1016I which is found all around Benin. As MCAs and AS-PCR are "leaky," 

For this revision, a caveat has been added to the discussion, but this same caveat is not applied at the level of the title or the abstract and both still claim, without reservation, that these alleles were found, when based on the actual data, the result is possibly found but unclear because there is no account for how 1016I appears in the 1016G assay. Most people will only read the title and abstract and thus never see the caveat but will come away thinking the alleles are found. 

The best resolution would have been to include the proper controls initially. Since this is not possible at this point, Sanger of these potential 1016G samples would be confirmatory and make the finding conclusive. If this is not possible because the samples are not extant, then the title and abstract need to make clear that this is a possible identification that requires additional work to confirm. Maybe change the title to: Insecticide resistance in Aedes aegypti mosquitoes: possible detection of kdr F1534C, S989P and V1016G triple mutation in Benin, West Africa

Author Response

Message well received

I was surprised by the absence of notification of my reply sent since 4 days.

No problem, the reviewer's only comment has been taken into account.

Round 4

Reviewer 2 Report (Previous Reviewer 2)

Comments and Suggestions for Authors

As in previous reviews, I still remain very concerned about publishing assays without positive controls for the mutant alleles, the lack of testing for 1016I since it is present in neighboring countries, and the lack of confirmatory testing for the finding of 1016G in the area but I appreciate that the Authors have at least acknowledged these issues.

If the Authors are comfortable publishing a surrogate assay without important controls or including a confirmatory method, then I will not continue to prevent it.

This manuscript is a resubmission of an earlier submission. The following is a list of the peer review reports and author responses from that submission.

Round 1

Reviewer 1 Report

Comments and Suggestions for Authors

The study conducted by Tokponnon et al. has highlighted the presence of insecticide-resistant Ae. aegypti mosquitoes in certain areas of Benin. They found that mosquitoes have developed resistance to permethrin and deltamethrin. However, they are still susceptible to bendiocarb. Through allele-specific PCR analysis of the Kdr gene, three kdr mutations were found to have high frequencies in Ae. aegypti populations in Benin regions. Furthermore, the mosquitoes present in Benin exhibit elevated levels of GSTs and decreased levels of esterases.

Most of the conclusions are well supported by the data although there are some things below the authors should address

1) A graph should be created to effectively showcase the insecticide resistance profiles of permethrin, deltamethrin, and bendicarb

2) It would be interesting to identify the specific kdr mutations or combinations responsible for permethrin or deltamethrin resistance. Authors can determine this information by conducting an allele-specific PCR on mosquitoes that have survived an insecticide-resistant bioassay.

3) The discussion is well-written, but it would be helpful to include how the findings of this study can be applied to develop vector control strategies in Benin.

Author Response

RESPONSE : We would like to thank the reviewers for their time and their constructive comments, that have helped us significantly improve our manuscript. Responses to all comments are given below. Please note line numbers refer to the « All Markup » tracked changes manuscript version.

The study conducted by Tokponnon et al. has highlighted the presence of insecticide-resistant Ae. aegypti mosquitoes in certain areas of Benin. They found that mosquitoes have developed resistance to permethrin and deltamethrin. However, they are still susceptible to bendiocarb. Through allele-specific PCR analysis of the Kdr gene, three kdr mutations were found to have high frequencies in Ae. aegypti populations in Benin regions. Furthermore, the mosquitoes present in Benin exhibit elevated levels of GSTs and decreased levels of esterases.

Most of the conclusions are well supported by the data although there are some things below the authors should address

RESPONSE : We thank the reviewer for their positive feedback.

  • A graph should be created to effectively showcase the insecticide resistance profiles of permethrin, deltamethrin, and bendicarb

RESPONSE : We have added the requested figure (now Figure 2).

2) It would be interesting to identify the specific kdr mutations or combinations responsible for permethrin or deltamethrin resistance. Authors can determine this information by conducting an allele-specific PCR on mosquitoes that have survived an insecticide-resistant bioassay.

RESPONSE : Table 1 has now been revised to include this information

  • The discussion is well-written, but it would be helpful to include how the findings of this study can be applied to develop vector control strategies in Benin.

RESPONSE : The discussion has been expanded to include this information (final paragraph)

Reviewer 2 Report

Comments and Suggestions for Authors

This manuscript examines levels of insecticide resistance to common active ingredients in two populations of Aedes aegypti from Benin. It also makes a preliminary assessment of the presence of biochemical and kdr mutations that may play a role in the noted resistance to pyrethroids. The reason for the study is important and the data set is interesting and should be published after a few significant issues are addressed that will allow a full evaluation of the results and conclusions. Specific issues to be addressed are noted by section below but the primary revisions needed fall into three areas. First, the introduction is extremely brief and could use a little more background, particularly as it relates to AI usage in Benin. Second, the methods are lacking adequate description in several areas but primarily with respect to controls (both for biochemical assays and kdr assays). Assays without proper controls are by default invalid so it makes evaluating the results and discussion impossible. Third, the statistical analysis used need to be much more thoroughly explained and whether the underlying assumptions of the tests are met.

As noted, this study has merit and should be published when the deficiencies are addressed. One thing puzzles me about the context of the results that should be considered in the next revision. Only minor phenotypic resistance is detected to permethrin and deltamethrin unlike the often strong resistance (<30% mortality at DD and DT) detected in the Americas and Asia. However, a high frequency of kdr mutations and enzymatic activity differences are detected. Isn't this counterintuitive for these to be frequent in a population of relatively low resistance? Taiwan CDC has shown the 989P & 1016G combination results in relatively strong resistance and that 1534C is not found on the same transcript as 989P/1016G. If your data is correct, it doesn't accord with the 989/1016 data from Asia because you show only minor resistance. The high frequency of 1534C is not so troubling. Several groups have shown it can be frequent even in moderately resistant populations. Without accounting for 1016I in your assays (which is often found in ensemble with 1534C in the Americas), it is hard to make sense of this data.

Specific comments by section:

The asterisk for the corresponding Author is not used. Also, Author Messenger is noted as 6,7 but no 7 is listed in the affiliations.

Simple summary: Please check capitalization of Gravid Aedes trap. Are you referring to the Gravid Aedes Trap (GAT) as below or is it being used here more generally? The capitalization will need to be adjusted depending upon usage.

Abstract: Suggest replacing "insecticide-susceptibility" with insecticide-susceptible.

Introduction: Suggest that the 2nd and 3rd sentences be combined into one. They are somewhat repetitive.

Introduction-2nd paragraph: Please provide references to support the statement made in the third sentence. Undoubtedly pressure from agriculture and malaria control drive IR in Anopheles but I can not off the top of my head think of references that show these reasons behind Aedes aegypti resistance. Dengue control, yes.

Please provide a reference for the WHO definition of resistance.

To make the introduction more complete, a discussion of the chemicals being used currently in Benin would be useful as well as the current kdr landscape in West Africa. Including this would more thoroughly develop the rationale for this study.

Section 2.2: Reference [17] does not go to the protocol but goes to a general malaria page. Please use: https://www.cdc.gov/malaria/resources/pdf/fsp/ir_manual/ir_cdc_bioassay_en.pdf

Also, were 3-4 bottles used for each AI as per the protocol. Were control bottles (acetone only) used as per the protocol? 

Note also that the dose of permethrin used in this study varies from the CDC protocol (permethrin=15ug/bottle, 30 min) and should thus be specifically noted along with the diagnostic times used for each AI.

Section 2.4: Please cite a reference for the basic protocol of these biochemical assays.

The non-specific esterase section is a little unclear. It seems something is missing. Ninety ul of 1%TBS is added to 10ul of lysate. Why is the diluted solution then incubated for 10 minutes? Biochemically, what is happening during the ten minutes? 

Also, please provide the final concentrations in the naphthyl/1%TBS/H2O solutions rather than describing how the solution is made. Same for the Fast Garnett Salt solution.

Was the reaction stopped with 5% SDS?

What controls were used so that the absorbance can be converted to umol? What negative controls were used?

MFO paragraph: Again, please describe the final concentration of the solutions added rather than the recipe. Note, the description of 25ul of 3% H2O2 is a good example of the suggested description. Also, what controls were used for the conversion from absorbance to nmol P450?

Same comments for GSTs and protein assays.

Section 2.5: This section needs copyedits and edits for clarity. Several issues need to be addressed.

First, has the AS-PCR method used here been published previously? If so, cite it. If not, you need to provide validation for your method.

Second, what controls were used in these assays? Were negative controls run? Did they give no signal? What allele specific positive controls were included? Did they give the expected signal?

Third, it is a little unclear how exactly the mutant and the wildtype are being differentiated for each SNP. Only 2 primers are listed for each SNP, a forward and reverse. How does this result in bands of two different sizes to differentiate the alleles?

Fourth, the 1016 SNP can result in a V, a G, or an I, depending on which position of the codon is mutated or not mutated. I believe 1016I has been reported in West Africa. How does this assay respond if 1016I is present rather than 1016G? These issues must be clearly explained. 

Fifth, provide the PCR recipe in final concentrations rather than the recipe. What vendors provided the Taq and the PCR buffer?

Section 2.6: How were bottle bioassay results analyzed using excel?

Specifically, how were biochemical assay results analyzed using PRISM? What statistical tests? ANOVA? Were the assumptions met?

Pearson's Chi squared assumes categorical data and is specifically stated as being used to observe deviations from HWE. HWE assumes a population under no selective pressure where two alleles exist in balance. IN such case, what is the expected frequency of a resistance mutation? Doesn't the presence of a resistance mutation in a population imply that the population is or has been under selective pressure? While you can certainly cite existing literature that has improperly applied HWE to IR SNPs, note that it violates the assumptions of HWE to do so. Suggest you do not fall into this error.

Section 3.2:  Surprising that both populations responded essentially the same to an equivalent amount of permethrin or deltamethrin. The data is not shown but it certainly should be. 

Section 3.3: It is unclear what statistical test was used for this analysis so this result is difficult to assess as valid or invalid. Note also, the study assesses the activity of these enzymes, not expression.

Section 3.4: Difficult to evaluate this without explanation of controls and specific analysis methods. See comments in methods.

Discussion: Can not evaluate the discussion unless the issues of methods and results are corrected.

Comments on the Quality of English Language

Minor edits needed but generally fine.

Author Response

RESPONSE : We would like to thank the reviewer for their time and their constructive comments, that have helped us significantly improve our manuscript. Responses to all comments are given below. Please note line numbers refer to the « All Markup » tracked changes manuscript version.

This manuscript examines levels of insecticide resistance to common active ingredients in two populations of Aedes aegypti from Benin. It also makes a preliminary assessment of the presence of biochemical and kdr mutations that may play a role in the noted resistance to pyrethroids. The reason for the study is important and the data set is interesting and should be published after a few significant issues are addressed that will allow a full evaluation of the results and conclusions. Specific issues to be addressed are noted by section below but the primary revisions needed fall into three areas.

First, the introduction is extremely brief and could use a little more background, particularly as it relates to AI usage in Benin.

RESPONSE : We have added information to the introduction about use of

Second, the methods are lacking adequate description in several areas but primarily with respect to controls (both for biochemical assays and kdr assays). Assays without proper controls are by default invalid so it makes evaluating the results and discussion impossible.

RESPONSE : We have now clarified the methods descriptions to address the comments below.

Third, the statistical analysis used need to be much more thoroughly explained and whether the underlying assumptions of the tests are met.

RESPONSE : We have now clarified the statistical analysis to address the comments below.

As noted, this study has merit and should be published when the deficiencies are addressed. One thing puzzles me about the context of the results that should be considered in the next revision. Only minor phenotypic resistance is detected to permethrin and deltamethrin unlike the often strong resistance (<30% mortality at DD and DT) detected in the Americas and Asia. However, a high frequency of kdr mutations and enzymatic activity differences are detected. Isn't this counterintuitive for these to be frequent in a population of relatively low resistance?

Taiwan CDC has shown the 989P & 1016G combination results in relatively strong resistance and that 1534C is not found on the same transcript as 989P/1016G. If your data is correct, it doesn't accord with the 989/1016 data from Asia because you show only minor resistance. The high frequency of 1534C is not so troubling. Several groups have shown it can be frequent even in moderately resistant populations. Without accounting for 1016I in your assays (which is often found in ensemble with 1534C in the Americas), it is hard to make sense of this data.

RESPONSE : We thank the reviewer for this insight. We believe that the occurrence of these mutations may be associated with a fitness cost, which may be adversely affecting mosquito survival following bioassay exposure. We have added this point to the discussion section.

Specific comments by section:

The asterisk for the corresponding Author is not used. Also, Author Messenger is noted as 6,7 but no 7 is listed in the affiliations.

RESPONSE : Amended accordingly.

Simple summary: Please check capitalization of Gravid Aedes trap. Are you referring to the Gravid Aedes Trap (GAT) as below or is it being used here more generally? The capitalization will need to be adjusted depending upon usage.

RESPONSE : We are referring to Gravid Aedes Trap (GAT) and have amended accordingly.

Abstract: Suggest replacing "insecticide-susceptibility" with insecticide-susceptible.

RESPONSE : Amended accordingly.

Introduction: Suggest that the 2nd and 3rd sentences be combined into one. They are somewhat repetitive.

RESPONSE : Amended accordingly.

Introduction-2nd paragraph: Please provide references to support the statement made in the third sentence. Undoubtedly pressure from agriculture and malaria control drive IR in Anopheles but I can not off the top of my head think of references that show these reasons behind Aedes aegypti resistance. Dengue control, yes.

RESPONSE : We agree with the reviewer and have modified this sentance to no longer imply that malaria control is driving IR in Ae. aegypti.

Please provide a reference for the WHO definition of resistance.

RESPONSE : Added accordingly.

To make the introduction more complete, a discussion of the chemicals being used currently in Benin would be useful as well as the current kdr landscape in West Africa. Including this would more thoroughly develop the rationale for this study.

RESPONSE : Added accordingly.

Section 2.2: Reference [17] does not go to the protocol but goes to a general malaria page. Please use: https://www.cdc.gov/malaria/resources/pdf/fsp/ir_manual/ir_cdc_bioassay_en.pdf

RESPONSE : Amended accordingly.

Also, were 3-4 bottles used for each AI as per the protocol. Were control bottles (acetone only) used as per the protocol? 

RESPONSE: Yes – this has been clarified in the methods.        

Note also that the dose of permethrin used in this study varies from the CDC protocol (permethrin=15ug/bottle, 30 min) and should thus be specifically noted along with the diagnostic times used for each AI.

RESPONSE: Yes – this has been clarified in the methods.         

Section 2.4: Please cite a reference for the basic protocol of these biochemical assays.

RESPONSE: We have added the following citation: Hemingway J. Insecticide resistance mechanisms (Field and laboratory manual). Geneva: World Health Organization. Pp. 1-39. 1998.

The non-specific esterase section is a little unclear. It seems something is missing. Ninety ul of 1%TBS is added to 10ul of lysate. Why is the diluted solution then incubated for 10 minutes? Biochemically, what is happening during the ten minutes? 

RESPONSE : Amended accordingly.

Also, please provide the final concentrations in the naphthyl/1%TBS/H2O solutions rather than describing how the solution is made. Same for the Fast Garnett Salt solution.

RESPONSE : Amended accordingly.

Was the reaction stopped with 5% SDS?

RESPONSE : Amended accordingly.

What controls were used so that the absorbance can be converted to umol? What negative controls were used?

RESPONSE : Amended accordingly.

MFO paragraph: Again, please describe the final concentration of the solutions added rather than the recipe. Note, the description of 25ul of 3% H2O2 is a good example of the suggested description. Also, what controls were used for the conversion from absorbance to nmol P450?

RESPONSE : Amended accordingly.

Section 2.5: This section needs copyedits and edits for clarity. Several issues need to be addressed.

First, has the AS-PCR method used here been published previously? If so, cite it. If not, you need to provide validation for your method.

RESPONSE: Yes this assay has been published in reference #12, which we have now cited.

Second, what controls were used in these assays? Were negative controls run? Did they give no signal? What allele specific positive controls were included? Did they give the expected signal?

RESPONSE: Yes negative controls were used and gave no signal, positive controls were also used

Third, it is a little unclear how exactly the mutant and the wildtype are being differentiated for each SNP. Only 2 primers are listed for each SNP, a forward and reverse. How does this result in bands of two different sizes to differentiate the alleles?

RESPONSE: Each individual mosquito was tested by AS-PCR twice, the first PCR used a primer specific to the susceptible wild-type and the second PCR used a primer specific to the mutant. The two primers are specific for each mutation and have same size. The migration of the PCR products is done separately, for example if for the same sample the susceptible wild-type allele appears as well as the mutant allele, then this mosquito is therefore homozygous

Fourth, the 1016 SNP can result in a V, a G, or an I, depending on which position of the codon is mutated or not mutated. I believe 1016I has been reported in West Africa. How does this assay respond if 1016I is present rather than 1016G? These issues must be clearly explained. 

RESPONSE: According to the manufacturer the reagent used is specific to the 1016G mutation

Fifth, provide the PCR recipe in final concentrations rather than the recipe. What vendors provided the Taq and the PCR buffer?

RESPONSE : Amended accordingly.

Section 2.6: How were bottle bioassay results analyzed using excel?

RESPONSE: Excel had been used to present the results in the form of graphs

Specifically, how were biochemical assay results analyzed using PRISM? What statistical tests? ANOVA? Were the assumptions met?

Pearson's Chi squared assumes categorical data and is specifically stated as being used to observe deviations from HWE. HWE assumes a population under no selective pressure where two alleles exist in balance. IN such case, what is the expected frequency of a resistance mutation? Doesn't the presence of a resistance mutation in a population imply that the population is or has been under selective pressure? While you can certainly cite existing literature that has improperly applied HWE to IR SNPs, note that it violates the assumptions of HWE to do so. Suggest you do not fall into this error.

RESPONSE : Mann Whitney test was chosen for comparison and all amended accordingly

Section 3.2:  Surprising that both populations responded essentially the same to an equivalent amount of permethrin or deltamethrin. The data is not shown but it certainly should be. 

RESPONSE : Amended accordingly

Section 3.3: It is unclear what statistical test was used for this analysis so this result is difficult to assess as valid or invalid. Note also, the study assesses the activity of these enzymes, not expression.

RESPONSE : Amended accordingly

Section 3.4: Difficult to evaluate this without explanation of controls and specific analysis methods. See comments in methods.

RESPONSE : Observations have been taken into account to facilitate better understanding.